# Disseminated Cryptococcosis Is a Common Finding among Human Immunodeficiency Virus-Infected Patients with Suspected Sepsis and Is Associated with Higher Mortality Rates

**DOI:** 10.3390/jof9080836

**Published:** 2023-08-09

**Authors:** Tafese Beyene Tufa, Hans Martin Orth, Tobias Wienemann, Bjoern-Erik Ole Jensen, Colin R. Mackenzie, David R. Boulware, Tom Luedde, Torsten Feldt

**Affiliations:** 1Department of Gastroenterology, Hepatology and Infectious Diseases, University Hospital and Medical Faculty of the Heinrich, Heine University, 40225 Düsseldorf, Germany; hansmartin.orth@med.uni-duesseldorf.de (H.M.O.); bjoern-erikole.jensen@med.uni-duesseldorf.de (B.-E.O.J.); tom.luedde@med.uni-duesseldorf.de (T.L.); 2Hirsch Institute of Tropical Medicine, Asella P.O. Box 04, Ethiopia; 3College of Health Sciences, Arsi University, Asella P.O. Box 04, Ethiopia; 4Institute of Medical Microbiology and Hospital Hygiene, University Hospital Duesseldorf, Medical Faculty, Heinrich Heine University, Universitätsstr. 1, 40225 Düsseldorf, Germany; tobias.wienemann@med.uni-duesseldorf.de (T.W.); colin.mackenzie@med.uni-duesseldorf.de (C.R.M.); 5Department of Medicine, University of Minnesota, Minneapolis, MN 55455, USA; boulw001@umn.edu

**Keywords:** cryptococcal infection, sepsis, qSOFA, blood culture, HIV, Africa

## Abstract

Cryptococcosis is the leading cause of death among people with HIV in Sub-Saharan Africa. The lack of optimum diagnoses and medications significantly impair the management of the disease. We investigated the burden of cryptococcosis and related mortality among people with HIV and suspected sepsis in Ethiopia. We conducted a prospective study at (1) Adama Hospital Medical College and (2) Asella Referral and Teaching Hospital from September 2019 to November 2020. We enrolled adult, HIV-infected patients presenting with suspected sepsis and assessed their 28-day survival rates. We performed blood cultures and cryptococcal antigen (CrAg) testing. In total, 82 participants were enrolled with a median age of 35 years, and 61% were female. Overall, eleven (13%) had positive CrAg tests, of which five grew *Cryptococcus* in blood cultures. Despite high-dose fluconazole (1200 mg/d) monotherapy being given to those with positive CrAg tests, the 28-day mortality was 64% (7/11), with mortality being significantly higher than among the CrAg-negative patients (9% (6/71); *p* < 0.001). Cryptococcosis was the leading cause of mortality among HIV-infected sepsis patients in this Ethiopian cohort. The CrAg screening of HIV-infected patients attending an emergency department can minimize the number of missed cryptococcosis cases irrespective of the CD4 T cell count and viral load. These findings warrant the need for a bundle approach for the diagnosis of HIV-infected persons presenting with sepsis in low- and middle-income countries.

## 1. Introduction

Cryptococcal infection is one of the leading causes of death among people with HIV in Sub-Saharan Africa [1,2], despite the expansion of antiretroviral therapy (ART) coverage [3]. Limited diagnostic and therapeutic options significantly impair the outcome of people with the disease in this region [3]. Point-of-care tests and an optimized antifungal therapy, as recommended by the World Health Organization (WHO) [4,5], have a major impact on the clinical outcome, but these are still lacking in most of African health facilities, including in Ethiopia.

However, the first-line recommended antifungals [4], a single high dose of liposomal amphotericin B and a 14-day course of flucytosine and fluconazole in the induction phase of the cryptococcal treatment, are lacking in most health facilities in Africa.

In addition to providing optimal antifungal therapy, performing frequent lumbar puncture (LP) techniques, and controlling their intracranial pressure based on the patient’s clinical needs have a significant impact on their survival [6]. However, in Africa and other resource-limited settings, manometers used to measure intracranial pressure and LP sets are usually missing. A study from Tanzania shows that measuring the CSF pressure with intravenous tubing sets may be an alternative in the absence of a manometer in resource-limited health facilities [6]. Furthermore, patients with a previously unknown cryptococcal infection and who had recently initiated ART have an increased risk of immune reconstitution inflammatory syndrome (IRIS) and death [7].

In a previous study, we reported that the quick Sequential Organ Failure Assessment (qSOFA) score shows an inadequate performance for the early detection of sepsis in our study centers [8]. Sepsis is defined as life-threatening organ dysfunction caused by a dysregulated immune response to an infection, and a qSOFA score of ≥2 points indicates sepsis [9,10,11]. However, minimal information is available on the value of the qSOFA to predict mortality or the presence of cryptococcal meningitis in HIV-infected patients.

Due to a lack of diagnostic options and epidemiological data, the timely initiation of adequate sepsis therapy remains challenging. This problem is aggravated among patients with advanced HIV disease because of the large spectrum of potential pathogens, including fungi or polymicrobial infections [12]. Therefore, irrespective of the CD4 T-cell count and viral load, we offered blood culture and cryptococcal antigen (CrAg) testing for all HIV-infected patients with suspected sepsis who were presented to an adult emergency department or admitted to medical wards. In this study, we assessed the value of the qSOFA score for the early identification of cryptococcal sepsis and the associated mortality of HIV-infected patients. In general, our objective was to investigate the burden of cryptococcal infection and the related mortality rate among HIV-infected patients with suspected sepsis in a prospective multicenter study in Ethiopia.

## 2. Materials and Methods

We conducted a prospective observational cohort study in Ethiopia at (1) Asella Referral and Teaching Hospital and (2) Adama Hospital Medical College from September 2019 to November 2020. Eligible participants had suspected sepsis according to the qSOFA score (score ≥ 1). We collected participants’ socio-demographic and clinical data at admission (baseline) and vital signs at the baseline and after 24 h by trained study personnel. Participants were followed for 28 days either in person or via a phone interview if the participant was discharged or transferred to another hospital.

### 2.1. Blood Culture

We collected 40 mL blood samples using blood culture bottles (BacT/ALERT^®^3D aerobic and anaerobic vials) and incubated them for up to five days (BacT/ALERT^®^3D, bioMérieux, Marcy-l’Étoile, France). If the instrument detected growth, we immediately performed Gram staining and subcultures on blood-, MacConkey- and chocolate-agar for bacteria, and Sabouraud dextrose agar for yeasts. Various biochemical tests, like catalase, coagulase, urease, oxidase, motility, indole, glucose and lactose fermentation, citrate utilization, gas, and H2S production tests, were performed to identify the pathogens on site. Then, the isolates were preserved at −81 °C in Microbank^®^ vials (Pro-Lab Diagnostics Inc., Toronto, Canada) and later exported to Düsseldorf, Germany, for confirmation of species identification and antifungal susceptibility testing. The species identification was confirmed by using the matrix-assisted laser desorption/ionization time obtained via flight (MALDI-TOF) mass spectrometry (Vitek^®^ MS, bioMérieux).

### 2.2. Minimum Inhibitory Concentration (MIC) Test

Confirmed samples of *Cryptococcus neoformans* were cultured at 35 °C, and antifungal susceptibility testing was performed with reading t at 48 h of incubation at 30 °C following the manufacturer’s instructions of micro dilution according to the EUCAST protocol.

### 2.3. Cryptococcal Antigen (CrAg) Test

In addition to blood culture diagnostics, we collected 5 mL of blood for additional point-of-care tests, including a CrAg lateral flow assay, which was approved by the U.S. Food and Drug Administration in 2011 (Immy, Norman, OK, USA), according to the manufacturer’s instructions [13].

### 2.4. Ethical Considerations

The institutional ethical review board of Arsi University in Asella, Ethiopia, the National Ethical Review Board of the Ethiopian Ministry of Science and Technology in Addis Ababa, Ethiopia, and the Ethics Committee of the Faculty of Medicine, Heinrich-Heine-University, Düsseldorf, Germany, approved of the study protocol. All patients provided written consent.

### 2.5. Statistical Analysis

Data were entered and analyzed using IBM SPSS Statistics version 26 (IBM Corp., Armonk, NY, USA). Fisher’s exact and Pearson’s chi-squared tests were used for the analysis of quantitative variables. We used descriptive statistics, calculated percentages, and medians to describe the demographic characteristics. Kaplan–Meier survival analysis was used to obtain the 28-day mortality rate. The differences were considered to be statistically significant at *p* < 0.05.

## 3. Results

### 3.1. Sociodemographic of Participants

In total, 82 HIV-infected participants were enrolled in the study. Out of the total, 56% (46) were on antiretroviral therapy (ART). The median age was 35 years (IQR: 27–40), and 63% were female. We calculated the qSOFA score of the participants, and 64 (78%) had a qSOFA score of ≥2 at the baseline, and 53 (65%) had a ≥2 qSOFA at 24 h after hospital admission (Table 1).

### 3.2. Cryptococcal Antigenemia

The prevalence of cryptococcal antigenemia was 13% (11/82), and these patients did not differ according to their ART status. The rate of CrAg positivity was 11% (4/36) in ART-naïve patients and 15% (7/46) among patients receiving ART (who knew their HIV serostatus and previously underwent ART with likely poor adherence). CrAg positivity did not differ by sex. Cryptococcal antigenemia was present in 14% (9/64) of patients with a qSOFA score ≥ 2 at presentation and 15% (8/53) among participants with qSOFA scores ≥ 2 at 24 h (Table 2).

### 3.3. Blood Culture

The overall blood culture positivity rate was 15% (12/82): *Cryptococcus neoformans* (*n* = 5), *Escherichia coli* (*n* = 3), *Salmonella* Typhi (*n* = 1), *Staphylococcus aureus* (*n* = 1), *Streptococcus pneumoniae* (*n* = 1), and *Enterococcus faecalis* (*n* = 1). The sensitivity of blood cultures for cryptococcosis was 46% (5/11) (Table 3). All patients who were blood-culture-positive for yeast of *Cryptococcus neoformans* were positive in the CrAg testing.

### 3.4. Minimum Inhibitory of Concentration (MIC)

Out of the five participants with *Cryptococcus* isolated from blood cultures, only one isolate (20%) had an elevated fluconazole MIC of 16 mg/L, which means it was considered to be not fully susceptible (Table 4). None of the isolates were resistant to the most commonly recommended antifungal agents of amphotericin B or flucytosine.

### 3.5. Participants’ Follow-up Outcomes

We assessed the 28-day mortality rate in comparison to the qSOFA score. The mortality rates of HIV-infected patients with baseline qSOFA scores of one, two, and three were 12%, 16%, and 25%, respectively. Although it was higher in the HIV-infected patients with a qSOFA score of three, the difference at the baseline was not statically significant (*p* = 0.567). However, the mortality rate in the HIV-infected patients with a qSOFA score of three at 24 h after admission was very high and statistically significant (*p* = 0.011). The 28-day mortality rate of the participants with a qSOFA score after 24 h of admission was 7% (Figure 1). The 28-day mortality rate among the HIV-infected patients with qSOFA score ≥ 2 at baseline was 17% (11/65,) and among those with a qSOFA score ≥ 2 at after 24 h of admission, it was 21% (11/53).

The overall 28-day mortality rate was 16% (13/82). Although all the CrAg-positive patients received high-dose fluconazole monotherapy (1200 mg/d), the mortality rate was 64% (7/11). This mortality rate was significantly higher in the CrAg-positive patients (64%) than it was in the CrAg-negative HIV-infected suspected sepsis patients (9% (6/71); *p* < 0.001). As shown in Figure 2, the majority of deaths among the CrAg-positive patients occurred in the initial 20 days.

## 4. Discussion

*Cryptococcus neoformans* has recently been ranked among the very critical pathogenic fungi [14], and cryptococcosis is a neglected killer among HIV-infected individuals in low- and middle-income countries. In this study, we found that in addition to the CrAg screening of HIV-infected persons with a low CD4 T cell count, per WHO’s recommendations for preemptive fluconazole therapy, the CrAg screening of HIV-infected patients attending an emergency outpatient or internal medicine department irrespective of their CD4 T cell count can prevent the under diagnosis of cryptococcosis and its associated mortality in resource-limited settings.

We reported a 15% blood culture positivity rate among the HIV-infected patients for suspected sepsis, and this figure is higher than our previous finding of a 5.4% bacterial blood culture positivity rate in febrile patients, in which most participants were from the same hospital [15]. The pathogen profile detected via the blood cultures is somewhat different from that of non-HIV-infected patients with suspected sepsis. In the current study, we isolated *C. neoformans* (*n* = 5), *E. coli* (*n* = 3), *Salmonella* Typhi (*n* = 1), *S. aureus* (*n* = 1), *S. pneumoniae* (*n* = 1), and *E. faecalis* (*n* = 1). With the exception of *E. coli* and *S. aureus*, the detected pathogens were commonly isolated in our previous study [8]. However, all pathogens are known to be typical causes of bloodstream infections in HIV-infected individuals in resource-limited settings, including *Mycobacterium tuberculosis*, which was not targeted in this study [16,17].

Another important point is that, while all the patients for whom the blood cultures were positive for *Cryptococcus neoformans* were also positive in the CrAg testing, the reverse is not true. This clearly showed us that the sensitivity of CrAg testing for the diagnosis of cryptococcosis is better than that of blood culture.

We reported a 28-day mortality rates of 21% among the HIV-infected patients with a qSOFA score ≥ 2 at 24 h upon admission and 7% among the patients with a qSOFA score < 2, respectively. This mortality rate is lower than our previous report from the same hospital among adult sepsis patients with a qSOFA score ≥2 [8]. However, the mortality rate was very high among the CrAg-positive patients treated with high-dose fluconazole monotherapy (1200 mg/d) when compared with that of the CrAg-negative HIV-infected suspected sepsis patients (64% vs. 9%); *p* < 0.001) (Figure 2). This finding is largely consistent with our earlier report that 68% of cryptococcal patients died within the first three months of undergoing fluconazole therapy at 1200 mg/d [18], and the recent report from Sierra Leone noted that 63% of CrAg-positive participants died within a month [19]. There is considerable evidence that high-dose fluconazole monotherapy is inadequate for cryptococcal meningitis treatment, but fluconazole is effective as a preemptive therapy in asymptomatic antigenemia patients or when no brain infection is involved [20,21].

Our results showed that a qSOFA score ≥2 in HIV-infected patients had no statically significant effect on the proportion of cryptococcal antigenemia both at the baseline and after 24 h. Thus, a higher qSOFA value at 24 h predicted the mortality of HIV-infected patients (Figure 1), but a higher qSOFA value at the baseline did not predict their mortality. Therefore, new modeling approaches should be developed to easily predict cryptococcal infections and their associated mortality, considering other factors such as the CD4 T-cell count-altered mental status, and intracranial pressure [22], or CrAg serum titers [23].

In resource-limited settings, applying the existing diagnostic and therapeutic options without delay is crucial in order to minimize the cryptococcal-associated mortality among HIV-infected individuals [24]. Several approaches exist to reduce the rate of cryptococcal mortality. One approach is to expand ART services and improve patients’ adherence, and thus, immune status to reduce their susceptibility to cryptococcal infections. We reported a higher prevalence of them among those undergoing ART in this study, which could be due to their poor adherence to ART. Another approach is the implementation of the current WHO recommendations which advocate for CrAg screening among patients with advanced HIV disease and preemptive fluconazole treatments for asymptomatic CrAg-positive individuals without delay [4]. There is strong evidence that CrAg-positive HIV-infected patients with a CrAg-negative CSF show no difference in mortality compared to that of the CrAg-negative subjects when treated with fluconazole alone when their plasma titers are ≤1:80 [5,20,25]. The other option we have pointed out in this study is to offer CrAg testing to all HIV-infected patients admitted to hospitals or newly HIV-diagnosed people, taking into account other common opportunistic infections. Then, an adequate antifungal treatment should be administered to the CrAg-positive patients, and the patients should be closely monitored. For instance, a multicounty study has shown that a single dose of liposomal amphotericin B at 10 mg/kg followed by a combination of 5-flucytosine and fluconazole is cost-effective and safe [26].

Few data are available on the resistance profile of *C. neoformans* yeasts in Africa. In this study, we reported that one of five isolates was resistant to fluconazole (see Table 4), although the sample size was too small. There were no isolates that were resistant to the first line recommended antifungal agents, amphotericin B and 5-flucytosine. However, in most African countries, fluconazole is the only antifungal available in many health facilities for cryptococcal meningitis treatment. The current treatment guideline in Ethiopia recommends a high dose of fluconazole (fluconazole 600 mg twice daily alone) for the induction phase as option A and also a lower dose of fluconazole for consolidation and maintenance therapy [27]. The emerging fluconazole resistance of *C. neoformans* and the high mortality rate of cryptococcal patients receiving fluconazole monotherapy [26] require timely action to make other recommended antifungals available.

Unlike bacteria, multidrug resistance among fungi is rare and not yet widespread, although the number of fungi resistant to antifungals is increasing from time to time [28]. Therefore, it is recommended to quickly introduce fungal diagnostics and susceptibility testing for isolates to combat the spread of fungal resistance and prevent the emergence of multidrug resistance like *Candida auris*.

In this study, we found no correlation of resistance between fluconazole and amphotericin B or 5-flucytosine for the five isolated yeasts, and this result is consistent with previous reports [29]. Therefore, for the effective treatment and prevention of resistant yeasts expansion, the use of antifungal combination therapy is strongly recommended for cryptococcal treatment.

Our study has some limitations. The sample size was too small to generalize the data, and we did not consider tuberculosis and cytomegalovirus, which are also common causes of sepsis and/or contributors to mortality among HIV-infected patients.

## 5. Conclusions

Cryptococcal infection was the leading cause of mortality among HIV-infected suspected sepsis patients in the study sites. With limited therapeutic options, fluconazole monotherapy being the only treatment currently available, the 28-day mortality rate was alarmingly high (64%). In this study, we found that a higher qSOFA score 24 h after admission was not related to CrAg-positivity, but predicted a high 28-day mortality rate among the HIV-infected patients. In addition to scaling up ART coverage in the country and the improvement of patients’ adherence to HIV therapy, the capacity for HIV testing should be strengthened to reduce the number of HIV-infected individuals who are diagnosed late. The CrAg screening of newly diagnosed HIV-infection- and HIV-positive people who are attending a hospital irrespective of their CD4 T cell count and viral load can minimize the number of missed cryptococcal infection cases in resource-limited settings. Expanding the fungal laboratory capacities in the country and performing susceptibility testing for pathogenic fungal isolates are very important to minimize the resistance to antifungals and to improve patients’ care. These findings warrant the need of a bundle approach for the diagnostics and management of HIV-infected patients presenting with sepsis in Africa, including CrAg testing and mycobacterial testing.

## Figures and Tables

**Figure 1 jof-09-00836-f001:**
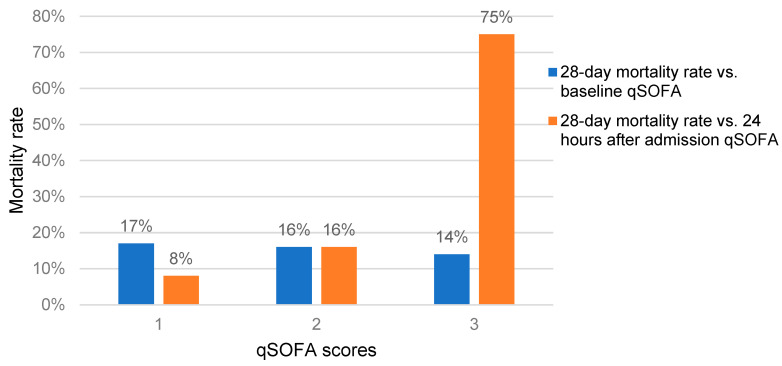
Twenty-eight-day mortality of HIV-infected patients with different qSOFA scores during admission as baseline and qSOFA scores at 24 h after of admission in two teaching hospitals in the Oromia region of Ethiopia.

**Figure 2 jof-09-00836-f002:**
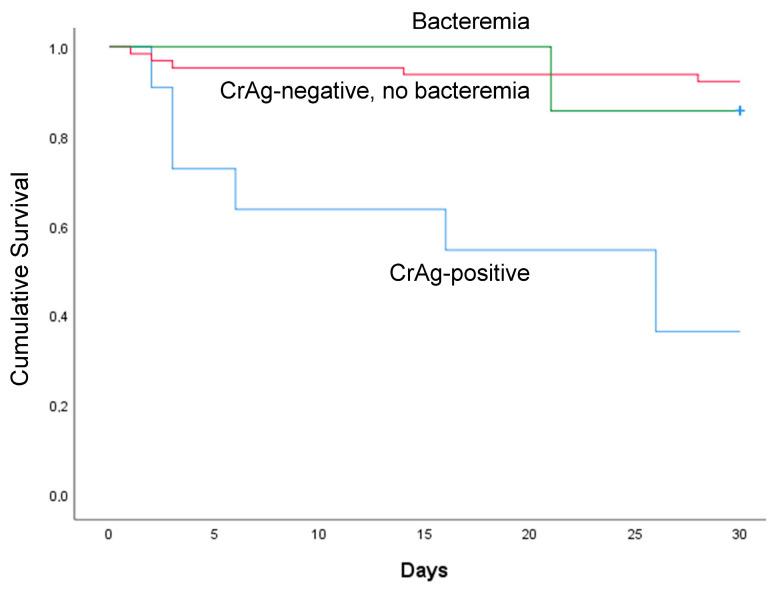
Kaplan–Meier survival analysis of 28 days of HIV-infected suspected sepsis patients at two teaching hospitals in Oromia region of Ethiopia.

**Table 1 jof-09-00836-t001:** Sociodemographic and clinical data of participants.

Variables	Frequency
Age in years, median (IQR)	35 (27–40)
Study site	
Asella Hospital	54 (66%)
Adama Hospital	28 (34%)
Sex	
Female	52 (63%)
Male	30 (37%)
HIV Therapy status	
ART-naïve	36 (44%)
Receiving ART	46 (56%)
qSOFA score	
≥2 at baseline	64 (78%)
≥2 after 24 h	53 (65%)

Values are N (%) or median (interquartile range). qSOFA: quick Sequential Organ Failure Assessment (qSOFA) score.

**Table 2 jof-09-00836-t002:** Cryptococcal antigen testing positivity rate among HIV-infected suspected sepsis patients.

Variables	Positive *n* (%)	Negative *n* (%)	Total *n* (%)	*p* Value
HIV Therapy status				
ART naïve	4 (11%)	32 (89%)	36 (44%)	
Receiving ART	7 (15%)	39 (85%)	46 (56%)	0.419
≥2 qSOFA score at baseline (***n*** = 64)	9 (14%)	55 (86%)	64 (78%)	0.549
≥2 qSOFA score after 24 h (***n*** = 53)	8 (15%)	45 (85%)	53 (65%)	0.406
Total	11 (13%)	71 (87%)	82 (100%)	

**Table 3 jof-09-00836-t003:** List of isolated pathogens from blood culture.

Name of the Microorganisms	Frequency of Isolates
*Cryptococcus neoformans*	5
*Escherichia coli*	3
*Salmonella t* *yphi*	1
*Staphylococcus aureus*	1
*Streptococcus pneumoniae*	1
*Enterococcus faecalis*	1
Total	12

**Table 4 jof-09-00836-t004:** Minimum inhibitory of concentration (MIC) distribution of C. neoformans isolates (*n*  =  5).

	Minimum Inhibitory of Concentration (mg/L)												Number Yeasts	
Antifungal Agents	≤0.015	0.03	0.06	0.12	0.25	0.5	1	2	4	8	16	>32	Susceptible	Resistant
Amphotericin B	0	0	0	0	2	3	0	0	0	0	0	0	5	0
5-Flucytosine	0	0	0	0	1	3	1	0	0	0	0	0	5	0
Fluconazole	0	0	0	0	2	2	0	0	0	0	1	0	4	1
Voriconazole	2	2	1	0	0	0	0	0	0	0	0	0	5	0
Posaconazole	0	4	1	0	0	0	0	0	0	0	0	0	5	0
Micafungin	0	0	0	0	0	0	0	0	0	1	4	0	0	5
Anidulafungin	0	0	0	0	0	0	0	0	0	0	5	0	0	5
Caspofungin	0	0	0	0	0	0	0	4	1	0	0	0	0	5

Fluconazole breakpoints for non-fully susceptible is ≥16 mg/L. The antifungal susceptibility testing result was interpreted according to FB-BA-057.06 EUCAST. Blue color is showing “no yeast or no color change” due to no growth, and the “red color” shows the presence of growth. “0” shows no new growth at the specified concentration and antifungal.

## Data Availability

The data presented in this study are available on request from the corresponding author. The data are not publicly available due to privacy of the patients.

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
