# Peer review of "Disseminated Cryptococcosis Is a Common Finding among Human Immunodeficiency Virus-Infected Patients with Suspected Sepsis and Is Associated with Higher Mortality Rates"

_jof, 2023, doi:10.3390/jof9080836_

Round 1

Reviewer 1 Report

In this manuscript, the authors evaluated the occurrence of cryptococcosis and related mortality among patients with HIV and sepsis in Ethiopia, in a prospective observational study. The authors discuss important aspects of patient care, diagnosis, and treatment, showing that cryptococcosis is the main cause of death in these patients. The paper is well written, and the aim of the study is important, given that this is a neglected disease, for which more information on mortality and morbidity is much needed.

Some minor changes should be made:

- "CD4 count" must be "CD4 T cell count"

- some microorganisms names should be in italic (some need review, line 193, for exemplo).  

- Line 201: “This mortality rate is lower than our previously reported figures”. This phrase seems incorrect. Would be it “our previous report”?

- Line 223: “s. Even though we reported higher prevalence of among on ART in this study, this could be due to their poor adherence for ART. Please, rewrite this phrase because it is confusing.

- Line 249: “Candida aureus” should be “Candida auris”.

In this manuscript, the authors evaluated the occurrence of cryptococcosis and related mortality among patients with HIV and sepsis in Ethiopia, in a prospective observational study. The authors discuss important aspects of patient care, diagnosis, and treatment, showing that cryptococcosis is the main cause of death in these patients. The paper is well written, and the aim of the study is important, given that this is a neglected disease, for which more information on mortality and morbidity is much needed.

Some minor changes should be made:

- "CD4 count" must be "CD4 T cell count"

- some microorganisms names should be in italic (some need review, line 193, for exemplo).  

- Line 201: “This mortality rate is lower than our previously reported figures”. This phrase seems incorrect. Would be it “our previous report”?

- Line 223: “s. Even though we reported higher prevalence of among on ART in this study, this could be due to their poor adherence for ART. Please, rewrite this phrase because it is confusing.

- Line 249: “Candida aureus” should be “Candida auris”.

Author Response

Dear Reviewer,

I would like to thank for taking your time and giving me very valuable comments and suggestions

Here I have addressed your comments point by point as follows

Reviewer 1

Point 1. "CD4 count" must be "CD4 T cell count":

Reply 1. Updated..

Point 2. some microorganisms names should be in italic (some need review, line 193, for exemplo):

Reply 2. corrected.

Point 3. Line 223: “s. Even though we reported higher prevalence of among on ART in this study, this could be due to their poor adherence for ART. Please, rewrite this phrase because it is confusing.

Reply 3. Corrected as “We reported higher prevalence of among on ART in this study, which could be due to their poor adherence for ART”

Point 4. Line 249: “Candida aureus” should be “Candida auris”.

Reply 4. Corrected.

Thank you very much, Tafese

Reviewer 2 Report

In this paper, the authors conducted a prospective cohort study in two hospitals in Ethiopia, from September 2019 to November 2020. The study included 82 HIV-infected patients with suspected sepsis in which blood cultures and cryptococcal antigen (CrAg) testing were performed. Among them, 11(13%) had positive CrAg tests, and 5 had Cryptococcus detected in blood cultures. The 28-day mortality was 64% (7/11) and was significantly higher in patients with positive CrAg tests than in those with negative CrAg tests: 9% (6/71); p < 0.001. The authors concluded that Cryptococcosis is the leading cause of mortality among HIV-infected sepsis patients and underlined that CrAg screening is useful in this population.

The authors are to be commended for their efforts to provide additional information on this pertinent question. However, this paper included too few patients, notably too few infected patients, to be able to draw any conclusions. In addition, other teams have already demonstrated the value of the cryptococcal antigen in HIV populations in Africa, and the effect of its positivity on mortality. 

For all these reasons, I don't think the manuscript can be published in a prestigious journal like JoF.

Minor comments: in the abstract: Limited diagnostic and therapeutic options significantly impair treatment options in Africa: please rephrase

Author Response

Dear Reviewer,

I would like to thank for taking your time and giving me very valuable comments and suggestions

Here I have addressed your comments point by point as follows

Reviewer 2

In this paper, the authors conducted a prospective cohort study in two hospitals in Ethiopia, from September 2019 to November 2020. The study included 82 HIV-infected patients with suspected sepsis in which blood cultures and cryptococcal antigen (CrAg) testing were performed. Among them, 11(13%) had positive CrAg tests, and 5 had Cryptococcus detected in blood cultures. The 28-day mortality was 64% (7/11) and was significantly higher in patients with positive CrAg tests than in those with negative CrAg tests: 9% (6/71); p < 0.001. The authors concluded that Cryptococcosis is the leading cause of mortality among HIV-infected sepsis patients and underlined that CrAg screening is useful in this population.

The authors are to be commended for their efforts to provide additional information on this pertinent question.

Point 1. this paper included too few patients, notably too few infected patients, to be able to draw any conclusions. In addition, other teams have already demonstrated the value of the cryptococcal antigen in HIV populations in Africa, and the effect of its positivity on mortality.

Reply 1.  We enrolled 82 hospitalized persons presenting with sepsis. This is a population which has not been extensively studied 1. Cryptococcal antigen screening for hospitalized populations or persons presenting with sepsis is not part of any hospital care guideline of which we are aware.

To raise awareness of the burden of cryptococcal infections in Ethiopia, we developed the manuscript as a communication that can go with this special issue, "Cryptococcosis and Cryptococcal Meningitis." We also know that many studies of cryptococcal antigen screening in HIV populations have been conducted in Africa, including the work of our team. However, data from Ethiopia are still limited. Moreover, this study should show that cryptococcal infection increases mortality in HIV-positive individuals compared with other pathogens. Therefore, we believe that it would be very helpful if our work were published as " communication" on JoF. If all HIV-positive patients coming to the emergency department are screened for CRAG regardless of CD4 count or VL, we can detect missing cases of cryptococcosis. This is clearly stated in the manuscript.

Point 2. in the abstract: Limited diagnostic and therapeutic options significantly impair treatment options in Africa: please rephrase

Reply 2. It is modfied as “Lack of optimum diagnosis and medications significantly impair the management of the disease”

Thank you very much, Tafese

1Moore CC, Jacob ST, Banura P, Zhang J, Stroup S, Boulware DR, Scheld WM, Houpt ER, Liu J. Etiology of Sepsis in Uganda Using a Quantitative Polymerase Chain Reaction-based TaqMan Array Card. Clin Infect Dis. 2019 Jan 7;68(2):266-272. doi: 10.1093/cid/ciy472. PMID: 29868873; PMCID: PMC6321855

Reviewer 3 Report

1. It remains to be specified what the criteria were to define sepsis.

2. Lines 60-61: However, the manometer to measure the intracranial pressure and the LP set are usually missing in this region.

In Africa and other regions of the world, CSF pressure is measured using intravenous tubing sets (Meda J, Kalluvya S, Downs JA, Chofle AA, Seni J, Kidenya B, Fitzgerald DW, Peck RN. Cryptococcal meningitis management in Tanzania with strict schedule of serial lumber punctures using intravenous tubing sets: an operational research study. J Acquir Immune Defic Syndr. 2014;66(2):e31-6. doi: 10.1097/QAI.0000000000000147).

3. I recommend updating reference 2:

Rajasingham R, Govender NP, Jordan A, Loyse A, Shroufi A, Denning DW, Meya DB, Chiller TM, Boulware DR. The global burden of HIV-associated cryptococcal infection in adults in 2020: a modelling analysis. Lancet Infect Dis. 2022;22(12):1748-1755. doi: 10.1016/S1473-3099(22)00499-6.

Author Response

Dear Reviewer,

I would like to thank for taking your time and giving me very valuable comments and suggestions

Here I have addressed your comments point by point as follows

Reviewer 3

Point 1. It remains to be specified what the criteria were to define sepsis:

Reply 1. We defined sepsis “Sepsis is defined as life-threatening organ dysfunction caused by a dysregulated im-munity response to an infection and a qSOFA score of ≥2 points indicates sepsis” (line 69-71)

Point 2. Lines 60-61: However, the manometer to measure the intracranial pressure and the LP set are usually missing in this region In Africa and other regions of the world, CSF pressure is measured using intravenous tubing sets.

Reply 3. We added the information (line 62-64).

Point 3. I recommend updating reference 2:

Rajasingham R, Govender NP, Jordan A, Loyse A, Shroufi A, Denning DW, Meya DB, Chiller TM, Boulware DR. The global burden of HIV-associated cryptococcal infection in adults in 2020: a modelling analysis. Lancet Infect Dis. 2022;22(12):1748-1755. doi: 10.1016/S1473-3099(22)00499-6..

Reply 3. updated.

with regards, Tafese

Reviewer 4 Report

This is an original article by a group of German/ Ethiopian authors who undertook a prospective study on causes of sepsis at 2 Eritrean hospitals- and this paper appears to reflect a substudy of causes in the HIV-infected subpopulation, whose stored blood samples underwent CrAg testing. It shows- like other African studies, how common CrAg-emia is in (hospitalised) patients with HIV (and therefore should be part of an screening bundle in this group), and the much higher mortality in CrAg+ patients. Whilst the implications of their findings are clear- the article would benefit from a more clear research aim - with a title and results and figures that reflect this. 

My suggestions:

Title- suggest reword to reflect the key findings - Disseminated cryptococcosis common finding in HIV patients with suspected sepsis and associated with higher mortality

Introduction- what is your paper's focus/ research question and hypothesis? Is it related to cryptococcal disease burden and contribution to mortality in this population- or is it related to the qSOFA score and its associations with mortality. To me it should be the former- especially given that you did not find association between qSOFA score and CRAG status, now qSOFA score and mortality (except at the later timepoint- which is not so relevant for this paper).

If you are going to give the qSOFA score- please define what you mean by sepsis or suspected sepsis- you say >=1- I looked it up and sepsis is >=2 plus suspected/ confirmed infection.  The score predicts mortality as opposed to diagnosing sepsis- and the latest survising sepsis guidelines do not recommend apparently https://www.mdcalc.com/calc/2654/qsofa-quick-sofa-score-sepsis#pearls-pitfalls. So you would need to justify how it is relevant to this particular research question. Given that CM is frequently associated with altered mental status these patients will automatically score 1 in the qSOFA score

Wording change- suggest not to use the term microbes but either causative organisms or pathogens

carefully check spelling of species names S pneumoniae, Candida auris

Methods

Was the CrAg assay done retrospectively and by whom and which method? Presumably serum CRAG not whole blood- please specify

State how patients were managed for CM- including antifungal treatment, LPs/ ICP management (if any)

You need a section on how statistical analyses were done eg comparison of Crag + and negative; mortality/survival analyses etc

Results-

As above I am unsure about use of qSOFA except just to report what it was- also you state >=2 qSOFA at baseline but >2 later - this reads a bit like data mining 

Did any of the CrAg positives have a history of prior CM?

It is relevant that 1/5 is fluco intermediate (do not use % when you are talking of single isolate) given that 1. this is current treatment in Ethiopia and 2. fluco is consolidation and maintenance therapy- you allude to this a bit more in the discussion

Discussion

Implementation- why recommend CrAg testing in suspected sepsis- why not any HIV-infected patient admitted? Or anyone with altered mental status? (given that CM is rarely associated with tachpneoa or hypotension) What is your local rate of HIV infection amongst medical admissions? 

I disagree that we need new models for mortality prediction- in crypto we have perfectly good (almost) bedside predictors using CrAg titer and Altered mental status- what we need in LMICs is better diagnostic algorithms and to do LPs

It is not correct to say there is good evidence that CRAG+ don't have excess mortality- other studies have shown an excess mortality in this group even when treated with fluconazole - hence the current EFFECT and ACACIA trials  testing intensified regimens for CrAg + patient (ask your co-author Prof Boulware)- you need to reference these studies (see Mfinanga S REMSTART study Lancet ID and Rachel Wake and Nicola Longley CID papers from South Africa)

Tables and figures

Only include those relevant to your main research question

Table 2- do not see how age or sex is relevant here- suggest omit

Table 3 not necessary

Table 4- simplify- do not show echinocandins as no activity. Use colours with traffic light system- with green indicating susceptible, red indicating resistant and yellow SDD

fig 1- wrong label- this is 28 day mortality rate not at 24h- suggest omit anyway as per my comments at the start

References

Many key studies on CRAG in Africa are not included- see above and please ask David Boulware to suggest additions

ref 4- ?? pamcdaa?

ref 23  is truncated

the article would benefit from some minor English language and grammar editing

Author Response

Dear Reviewer,

I would like to thank for taking your time and giving me very valuable comments and suggestions

Here I have addressed your comments point by point as follows

Reviewer 4

This is an original article by a group of German/ Ethiopian authors who undertook a prospective study on causes of sepsis at 2 Eritrean hospitals- and this paper appears to reflect a substudy of causes in the HIV-infected subpopulation, whose stored blood samples underwent CrAg testing. It shows- like other African studies, how common CrAg-emia is in (hospitalised) patients with HIV (and therefore should be part of an screening bundle in this group), and the much higher mortality in CrAg+ patients. Whilst the implications of their findings are clear- the article would benefit from a more clear research aim - with a title and results and figures that reflect this. 

My suggestions:

Point 1. Title- suggest reword to reflect the key findings - Disseminated cryptococcosis common finding in HIV patients with suspected sepsis and associated with higher mortality:

Reply 1. The title is corrected according to your suggestion.

Point 2. Introduction- what is your paper's focus/ research question and hypothesis? Is it related to cryptococcal disease burden and contribution to mortality in this population- or is it related to the qSOFA score and its associations with mortality. To me it should be the former- especially given that you did not find association between qSOFA score and CRAG status, now qSOFA score and mortality (except at the later timepoint- which is not so relevant for this paper):

Reply 2. Our main research question is to determine the major contribution factors for mortality among HIV positive sepsis suspected patients at the study sites since we found higher mortality among HIV positive participants during our main sepsis study. We have done a study to show the role of qSOFA scores to predict the mortality among sepsis patients at the study site before and we found high qSOFA scores had significant association with mortality in general sepsis patients. This is a reason why we also interested to discuss about qSOFA to recolonize cryptococcal infection associated sepsis and predict mortality among HIV-infected populations. Yes, it is to determine the cryptococcal disease burden and contribution to mortality in this population. 

Point 3. If you are going to give the qSOFA score- please define what you mean by sepsis or suspected sepsis- you say >=1- I looked it up and sepsis is >=2 plus suspected/ confirmed infection.  The score predicts mortality as opposed to diagnosing sepsis- and the latest surviving sepsis guidelines do not recommend apparently https://www.mdcalc.com/calc/2654/qsofa-quick-sofa-score-sepsis#pearls-pitfalls. So you would need to justify how it is relevant to this particular research question. Given that CM is frequently associated with altered mental status these patients will automatically score 1 in the qSOFA score

Reply 3. Yes, patient who has altered mental status will automatically score 1 in the qSOFA calculation. That is a reason that we used suspected sepsis in steady of sepsis. We included infection suspected HIV-positive patients who had 1 and above qSOFA score. If we only included patients who had >=2 plus suspected, we can use sepsis, not suspected sepsis.

Point 4. Wording change- suggest not to use the term microbes but either causative organisms or pathogens

Reply 4. Corrected.

Point 5. Carefully check spelling of species names S pneumoniae, Candida auris

Reply 5. Corrected.

Point 6. Was the CrAg assay done retrospectively and by whom and which method? Presumably serum CRAG not whole blood- please specify

Reply 6.  No, the CrAg assay was prospectively performed as part of screening from plasma of HIV-infected patients admitted to emergency department with suspected infections during the study period. This brief communication is a subset of those with sepsis.

Point 7. State how patients were managed for CM- including antifungal treatment, LPs/ ICP management (if any)

Reply 7. As patients were prospectively screened, all received the available antifungal therapy of fluconazole 1200mg/day. There other management was based on expert or treating clinicians’ opinion.

Point 8. You need a section on how statistical analyses were done eg comparison of Crag + and negative; mortality/survival analyses etc

Reply 8. The statistical analysis section was added (lines 121 -126)

Point 9. As above I am unsure about use of qSOFA except just to report what it was- also you state >=2 qSOFA at baseline but >2 later - this reads a bit like data mining 

Reply 9. The correct figure is ≥ 2.

Point 10. Did any of the CrAg positives have a history of prior CM?

Reply 10. Sorry we don’t have information about a history of prior CM. (As the mortality is very high with fluconazole monotherapy, we doubt there are prior survivors of CM where their CrAg-positivity is residual from the prior successfully treated infection. Successful treatment of infections is rare.

 Point 11. It is relevant that 1/5 is fluco intermediate (do not use % when you are talking of single isolate) given that 1. this is current treatment in Ethiopia and 2. fluco is consolidation and maintenance therapy- you allude to this a bit more in the discussion

Reply 11, Added and Corrected now (Line 255-257)

Point 12. Implementation- why recommend CrAg testing in suspected sepsis- why not any HIV-infected patient admitted? Or anyone with altered mental status? (given that CM is rarely associated with tachpneoa or hypotension) What is your local rate of HIV infection amongst medical admissions? 

Reply 12. We have corrected to all “hospital admitted HIV-infected patients.”

Point 13. I disagree that we need new models for mortality prediction- in crypto we have perfectly good (almost) bedside predictors using CrAg titer and Altered mental status- what we need in LMICs is better diagnostic algorithms and to do LPs

Reply 13. This study is a study of those presenting with sepsis, not meningitis. We know that higher CrAg titer can predict mortality and altered mental status has one point during qSOFA calculation. However, even though our sample size is too small we understood higher qSOFA scores which predict mortality in other sepsis cases, here is not statically significant. That why we propose new models rather than qSOFA score which can predict mortality in case of disseminated cryptococcosis in addition to having better diagnostic algorithms and performing serial LPs.

Point 14. It is not correct to say there is good evidence that CRAG+ don't have excess mortality- other studies have shown an excess mortality in this group even when treated with fluconazole - hence the current EFFECT and ACACIA trials testing intensified regimens for CrAg + patient (ask your co-author Prof Boulware)- you need to reference these studies (see Mfinanga S REMSTART study Lancet ID and Rachel Wake and Nicola Longley CID papers from South Africa).

Reply 14.  The reviewer is referencing this sentence in Line 247-249: “There is strong evidence that CrAg-positive HIV-infected patients with CrAg-negative CSF show no difference in mortality compared to CrAg-negative subjects when treated with fluconazole alone.” which is one conclusion that one would draw from the Longley CID 2016 study, and some guidelines which emphasize performing an LP on all subjects; and if CSF CrAg negative, then giving fluconazole.  Yes, we believe those guidelines are in error, and that CrAg titer is a better way to detect disseminated cryptococcosis which involves the CNS (including the parenchyma of the brain when the CSF is negative). While that is a discussion for a different time, we will improve the sentence in question by adding “…when plasma titers are <=1:80.”

Point 15. Table 2- do not see how age or sex is relevant here- suggest omit

Reply 15. We have removed age and sex.

Pont 16. Table 3 not necessary

Reply 16. We used this table to simple show the list of pathogens isolated from Blood culture. it is helpfully for the readers to understand easily.

Pont 17. Table 4- simplify- do not show echinocandins as no activity. Use colours with traffic light system- with green indicating susceptible, red indicating resistant and yellow SDD.

Reply 17. It is Minimum inhibitory of concentration (MIC) test. The color change from original to red shows the presence of growth in each cells and the number written in the table from 0 to 5 shows at which minimum concentration the yeast grow. I think this is more easily understandable. 

Point 18. Fig 1- wrong label- this is 28 day mortality rate not at 24h- suggest omit anyway as per my comments at the start

Reply 18. It is corrected as “28-day mortality of HIV-infected patients with different qSOFA scores at admission and 24 hours after admission in two teaching hospitals of Oromia region in Ethiopia”. This means we have calculated qSOFA scores two times at (we collect the fetal signs at admission plus after 24 hours the patient admitted in hospital.

Point 19. Many key studies on CRAG in Africa are not included- see above and please ask David Boulware to suggest additions

Reply 19. Thank you very much. We have used key studies we thought support our study. We have added additional references.

Point 20. ref 4- ?? pamcdaa?

Reply 20. Here corrected as “World Health Organization. ‎2022‎. Guidelines for diagnosing, preventing and managing cryptococcal disease among adults, adolescents and children living with HIV. World Health Organization. https://apps.who.int/iris/handle/10665/357088. License: CC BY-NC-SA 3.0 IGO”

Point 21. ref 23 is truncated

Reply 21. Here corrected as

Gerlach ES, Altamirano S, Yoder JM, Luggya TS, Akampurira A, Meya DB, Boulware DR, Rhein J, Nielsen K.

  1. ATI-2307 Exhibits Equivalent Antifungal Activity in Cryptococcus neoformans Clinical Isolates With High and Low Fluconazole IC (50)." Front Cell Infect Microbiol 11: 695240.

Tafese

Round 2

Reviewer 2 Report

Thanks to the authors for answering my queries. I'm still a little skeptical about the importance of this data, even if I understand the interest put forward by the authors. 

Author Response

Dear Reviewer,

Thank you very much for your time and concern. 

Point 1. Thanks to the authors for answering my queries. I'm still a little skeptical about the importance of this data, even if I understand the interest put forward by the authors.

Reply to point 1

This finding is very important for the management of CM in countries where CrAg screening is not yet fully implemented in HIV-infected individuals. Our results show how missing cryptococcusis can be minimized. The other point that makes our study very important is that we also diagnosed other pathogens and demonstrated the burden of C. neoformans in HIV-positive sepsis patients.

Tafese

Reviewer 4 Report

no further comments

as previously

Author Response

Dear Reviewer,

I would like to thank you for taking the time to provide us with very valuable comments.
Point 1. minor revision of English language needed.
Response to point 1
The manuscript has been reviewed again by a native English speaker.

Thank you, Tafese